# Thera: Aliasing-Free Arbitrary-Scale Super-Resolution with Neural Heat Fields

**Alexander Becker**[*]                                                      *alexander.becker@geod.baug.ethz.ch*
*Photogrammetry and Remote Sensing, ETH Zurich*

**Rodrigo Caye Daudt**[*]                                                  *rodrigo.cayedaudt@geod.baug.ethz.ch*
*Photogrammetry and Remote Sensing, ETH Zurich*

**Dominik Narnhofer**                                                    *dominik.narnhofer@geod.baug.ethz.ch*
*Photogrammetry and Remote Sensing, ETH Zurich*

**Torben Peters**                                                              *torben.peters@geod.baug.ethz.ch*
*Photogrammetry and Remote Sensing, ETH Zurich*

**Nando Metzger**                                                          *nando.metzger@geod.baug.ethz.ch*
*Photogrammetry and Remote Sensing, ETH Zurich*

**Jan Dirk Wegner**                                                              *jandirk.wegner@uzh.ch*
*Department of Mathematical Modeling and Machine Learning, University of Zurich*

**Konrad Schindler**                                                                    *schindler@ethz.ch*
*Photogrammetry and Remote Sensing, ETH Zurich*

**Reviewed on OpenReview:** *https://openreview.net/forum?id=GU8YOfmqyg*

## Abstract

Recent approaches to arbitrary-scale single image super-resolution (ASR) use neural fields to represent continuous signals that can be sampled at arbitrary resolutions. However, point-wise queries of neural fields do not naturally match the point spread function (PSF) of pixels, which may cause aliasing in the super-resolved image. Existing methods attempt to mitigate this by approximating an integral version of the field at each scaling factor, compromising both fidelity and generalization. In this work, we introduce neural heat fields, a novel neural field formulation that inherently models a physically exact PSF. Our formulation enables analytically correct anti-aliasing at any desired output resolution, and – unlike supersampling – at no additional cost. Building on this foundation, we propose *Thera*, an end-to-end ASR method that substantially outperforms existing approaches, while being more parameter-efficient and offering strong theoretical guarantees. The project page is at https://therasr.github.io.

## 1 Introduction

Over the years, learning-based image super-resolution (SR) methods have achieved increasingly better results. However, unlike interpolation techniques that can resample images at any resolution, these methods typically require retraining for each scaling factor. Recently, arbitrary-scale SR (ASR) approaches have emerged, which allow users to specify any desired scaling factor without retraining, significantly increasing flexibility (Hu et al., 2019). Notably, with LIIF, Chen et al. (2021) pioneered the use of neural fields for single-image SR, exploiting their continuous representation to enable SR at arbitrary scaling factors. LIIF has since inspired

---

[*]Equal contribution.

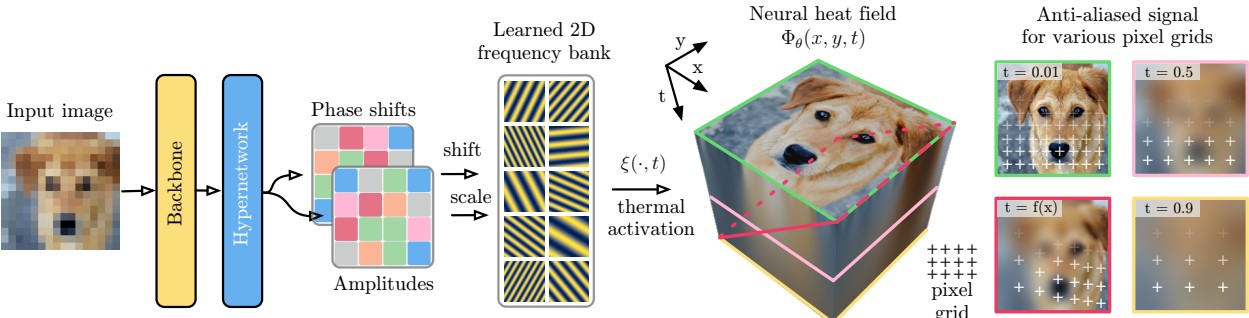

Figure 1: **We present *Thera*, the first method for arbitrary-scale super-resolution with a built-in physical observation model.** Given an input image, a hypernetwork predicts the parameters of a specially designed *neural heat field*, inherently decomposing the image into sinusoidal components. The field's architecture automatically attenuates frequencies as a function of the scaling factor so as to match the output resolution at which the signal is re-sampled.

several follow-ups which build upon the idea of using per-pixel neural fields (Lee & Jin, 2022; Cao et al., 2023; Chen et al., 2023; Zhu et al., 2025). This is not surprising: Neural fields are in many ways a natural match for variable-resolution computer vision and graphics (Xie et al., 2022). By implicitly parameterizing a target signal as a neural network that maps coordinates to signal value, they offer a compact representation, defined over a continuous input domain, and are analytically differentiable.

While neural fields naturally model continuous functions, they do not easily allow for observations of such functions other than point-wise evaluations. For many tasks, however, integral observation models such as point spread functions (PSFs) are desirable. This is particularly true for neural fields-based ASR methods, which by nature do not commit to a fixed upscaling factor *a priori* but regress continuous representations with unbounded spectra that can be observed at various sampling rates. If the Nyquist frequency corresponding to the desired sampling rate is lower than the highest frequency represented by the field, the sampling operation is prone to aliasing. This explains the initially counterintuitive relevance of anti-aliasing for super-resolution: When using neural fields, signals are first upsampled to *infinite* (continuous) resolution and then resampled at the desired resolution, and this latter operation must be done carefully. Incorporating a physically plausible observation model is not trivial (Barron et al., 2021; 2022; Lindell et al., 2022; Yang et al., 2022; Hu et al., 2023; Barron et al., 2023), but has the potential to avoid aliasing. For this reason, Chen et al. (2021) and successor works (Lee & Jin, 2022; Cao et al., 2023; Chen et al., 2023; Zhu et al., 2025) have already taken a first step towards learning multi-scale representations, via cell encoding. Fundamentally, these "learning-based anti-aliasing" approaches require the scaling factor (or, equivalently, the output pixel area) as additional input to the neural field and learn an integrated (*i.e.*, appropriately blurred and therefore anti-aliased) version of the field for each scaling factor; arguably wasting field capacity to approximate a relation that can be described exactly through Fourier theory.

In this work, we combine recent advances in implicit neural representations with ideas from classical signal theory to introduce *neural heat fields*, a novel type of neural field that *guarantees anti-aliasing by construction*. The key insight is that sinusoidal activation functions (Sitzmann et al., 2020b) enable selective attenuation of individual components depending on their spatial frequency, following Fourier theory. This allows for the exact computation of Gaussian-blurred versions of the field for any desired (isotropic) blur radius. When rasterizing an image, the field can therefore be queried with a Gaussian PSF that matches the target resolution, effectively preventing aliasing. In practice, heat fields receive an additional input coordinate $t$, controlling the strength of the Gaussian blur applied to the signal. Unlike learning-based anti-aliasing, the resulting filtering operation is expressed analytically, rather than learned from data. In other words, previous approaches fit a 3D field ($x$, $y$, and *scale*) while we only need to fit a 2D field ($x$ and $y$), whereas the scale dimension is computed analytically, significantly reducing field complexity and data requirements. Notably,

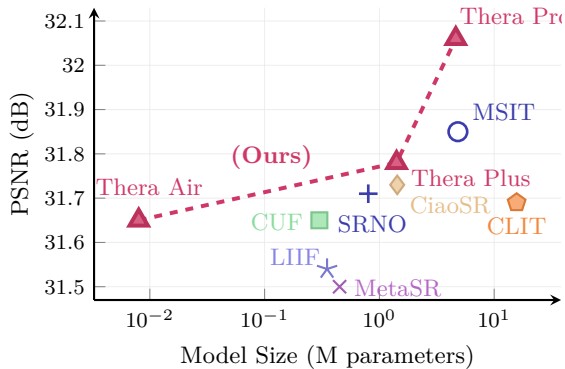

Figure 2: Comparison of recent ASR methods, averaged over $\times\{2, 3, 4\}$ scales. We generally achieve higher performance at lower parameter counts. Our best model, *Thera Pro*, achieves highest overall performance by a large margin.

filtering with neural heat fields incurs no computational overhead: The querying cost is the same for any width of the anti-aliasing filter kernel, including infinite and zero widths.

Building on this, we then propose *Thera*, an end-to-end ASR method that combines a hypernetwork (Ha et al., 2017) with a grid of local neural heat fields, offering theoretical guarantees with respect to multi-scale representation (see Figure 1). Empirically, *Thera* outperforms all competing ASR methods, often by a substantial margin, and is more parameter-efficient (see Figure 2). To the best of our knowledge, *Thera* is also the first neural field method to allow bandwidth control at test time.

In summary, our main contributions are:

1. We introduce neural heat fields, which represent a signal with a built-in, principled Gaussian observation model, and therefore allow anti-aliasing with minimal overhead.

2. We use neural heat fields to build *Thera*, a novel method for ASR that offers theoretically guaranteed multi-scale capabilities, delivers state-of-the-art performance and is more parameter efficient than prior art.

## 2 Related Work

### 2.1 Neural Fields

A neural field, also called an *implicit neural representation*, is a neural network trained to map coordinates onto values of some physical quantity. Recently, neural fields have been used for parameterizing various types of visual data, including images (Karras et al., 2021; Sitzmann et al., 2020b; Tancik et al., 2020; Chen et al., 2021; Lee & Jin, 2022; de Lutio et al., 2019; Wu et al., 2023), 3D scenes (*e.g.*, represented as signed distance fields (Park et al., 2019; Sitzmann et al., 2020b;a; Williams et al., 2022; Wu et al., 2023), occupancy fields (Mescheder et al., 2019; Peng et al., 2020), LiDAR fields (Huang et al., 2023), view-dependent radiance fields (Mildenhall et al., 2021; Barron et al., 2021; 2022; 2023; Wu et al., 2023)), or digital humans (Yenamandra et al., 2021; Zheng et al., 2022; Cao et al., 2022; Xiu et al., 2022; Giebenhain et al., 2023). Frequently, it is desirable to impose some prior over the space of learnable implicit representations. A common approach for such conditioning is encoder-based inference (Xie et al., 2022), where a parametric encoder maps input observations to a set of latent codes $z$, which are often local (Chen et al., 2021; Lee & Jin, 2022; Vasconcelos et al., 2023; Cao et al., 2023; Chen et al., 2023). The encoded latent variables $z$ are then used to condition the neural field, for instance by concatenating $z$ to the coordinate inputs or through a more expressive hypernetwork (Ha et al., 2017), mapping latent codes $z$ to neural field parameters $\theta$. An early example of this approach, which is gaining popularity (Xie et al., 2022), was proposed in Sitzmann et al. (2020b).

## 2.2 Arbitrary-Scale Super-Resolution

ASR is the sub-field of single-image SR in which the desired SR scaling factor can be chosen at inference time to be (theoretically) any positive number, allowing maximum flexibility, such as that of interpolation methods. The first work along this line is MetaSR (Hu et al., 2019), which infers the parameters of a convolutional upsampling layer using a hypernetwork conditioned on the desired scaling factor. An influential successor work is LIIF (Chen et al., 2021), in which the high-resolution image is implicitly described by local neural fields. These fields are conditioned via concatenation, with features extracted from the low-resolution input image. The continuous nature of the neural fields allows for sampling target pixels at arbitrary locations and thus also arbitrary resolution.

Most subsequent work has since been built upon the LIIF framework. For example, UltraSR (Xu et al., 2021) improves the modeling of high-frequency textures with periodic positional encodings of the coordinate space, as is common practice for *e.g.*, neural radiance fields (Mildenhall et al., 2021; Barron et al., 2021; 2022; 2023). LTE (Lee & Jin, 2022) makes learning higher frequencies more explicit by effectively implementing a learnable coordinate transformation into 2D Fourier space, prior to a forward pass through an MLP. Vasconcelos et al. (2023) use neural fields in CUF to parameterize continuous upsampling filters, which enables arbitrary-scale upsampling. More recently, methods like CiaoSR (Cao et al., 2023), CLIT (Chen et al., 2023), and most recently MSIT (Zhu et al., 2025) have integrated (multi-scale) attention mechanisms, improving reconstruction quality. In a parallel line of research, Wei & Zhang (2023) propose SRNO, an attention-based neural operator that learns a continuous mapping between low- and high-resolution function spaces.

Another line of work employs generative models such as denoising diffusion for SR (Saharia et al., 2022; Gao et al., 2023). While most methods minimize per-pixel errors (essentially predicting the minimum mean square error estimate), generative models are trained to produce more realistic-looking outputs by predicting one of many plausible high-resolution images. However, since specific ground truth details are not exactly recovered, such models typically report worse distortion metrics (like PSNR and SSIM) compared to pixel-based methods, c.f. Blau & Michaeli (2018); Delbracio & Milanfar (2023). In this paper, we adopt a pixel-based objective to preserve fidelity to the ground truth, which is important for many downstream applications (*e.g.*, face or license plate recognition).

## 2.3 Anti-Aliasing in Neural Fields

Early in the recent development of implicit neural representations, concerns regarding aliasing were raised. Barron et al. (2021) proposed integrating a positional encoding with Gaussian weights, which reduced aliasing in NeRF (Mildenhall et al., 2021). Improvements were later proposed for unbounded scenes (Barron et al., 2022) and to improve efficiency (Hu et al., 2023). Barron et al. (2023) tackle anti-aliasing within the Instant-NGP (Müller et al., 2022) approach. Recent work has succeeded in limiting the bandwidth using multiplicative filter networks (Lindell et al., 2022), polynomial neural fields (Yang et al., 2022) or cascaded training (Shabanov et al., 2024), although these works are restricted to discrete, pre-defined band limits (and thus resolutions) and have not tackled super-resolution tasks. These methods are not a good fit for ASR because they do not allow for continuous anti-aliasing, nor bandwidth control at test time. To perform scale-dependent filtering, most fields-based ASR methods instead explicitly provide the scale as input to the field, attempting to learn an appropriate observation model from data. While this approach may work reasonably well in in-distribution settings, it seeks to learn a model from data that can be described exactly with a differential equation, ultimately sacrificing fidelity and generalization.

In contrast, in this paper we explore a way to directly integrate a physics-informed observation model into the neural field representation.

# 3 Method

In this section we introduce *Thera*, a novel neural fields-based ASR method that guarantees analytical anti-aliasing at any desired output resolution at no additional cost. First, we present *neural heat fields*, a special

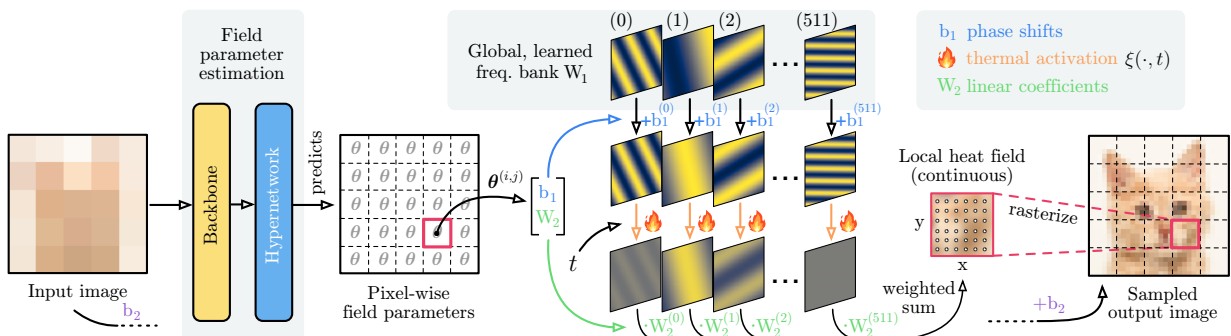

Figure 3: Overview of *Thera*. A hypernetwork estimates parameters $\{\boldsymbol{b}_1, \boldsymbol{W}_2\}^{(i,j)}$ of pixel-wise, local neural heat fields. The phase shifts $\boldsymbol{b}_1$ operate on globally learned components, before thermal activations scale each component depending on their frequency and the desired scaling factor. The components are then linearly combined using coefficients $\boldsymbol{W}_2$, resulting in an appropriately-blurred, continuous local neural field. This field is then rasterized at the appropriate sampling rate (resolution) to yield a part of the final output image (red square).

type of neural field that inherently achieves anti-aliasing by implicitly attenuating high-frequency components as a function of a time coordinate. Next, we propose a mechanism for learning a prior over a grid of neural heat fields, enabling them to represent a multi-scale output image conditioned on a lower-resolution input image. Finally, we show that our formulation allows us to impose a regularizer on the underlying, continuous signal itself – something that, to the best of our knowledge is not possible in previous methods.

## 3.1 Neural Heat Fields for Analytical Anti-Aliasing

Let $\mathbf{x} \in \mathbb{R}^2$ denote the spatial coordinates of a continuous image function $f(\mathbf{x})$. Aliasing occurs when this continuous signal is sampled at a rate that does not adequately capture its highest frequency components, resulting in overlapping spectral replicas in the Fourier domain. One must therefore apply a low-pass filter $g(\mathbf{x})$ whose cut-off frequency is aligned with the Nyquist frequency of the sampling rate, then sample the band-limited signal $f \circledast g(\mathbf{x})$. The key of our method is that, if a signal is decomposed into sinusoidal components, such filtering can be done simply by re-scaling each component by a factor that depends on their frequency as well as a time coordinate, which we call $t$. The time coordinate acts as a third, continuous input to the neural field and controls the amount of re-scaling, and therefore the Gaussian blur applied to the signal. This perfectly mimics how high-frequency components decay faster than low-frequency ones in the analytical solution to the heat equation. The detailed derivation can be found in Appendix A. The behavior described above is naturally accomplished by parameterizing the field $\Phi$ as a two-layer perceptron,

$$\Phi(\mathbf{x}, t) = \mathbf{W}_2 \cdot \xi\left(\mathbf{W}_1 \mathbf{x} + \mathbf{b}_1, \nu(\mathbf{W}_1), \kappa, t\right) + \mathbf{b}_2, \tag{1}$$

with parameters $\boldsymbol{\theta} := \{\mathbf{W}_1, \mathbf{W}_2, \mathbf{b}_1, \mathbf{b}_2\}$. Intuitively, $\mathbf{W}_1$ serves as a frequency bank, with its components acting as the basis functions that compose the signal $\Phi(\mathbf{x}, 0)$, and phase shifts encoded by $\mathbf{b}_1$. The matrix $\mathbf{W}_2$, with one row per output channel, contains initial magnitudes of these components, and $\mathbf{b}_2$ is the global bias of $\Phi$ per channel. Finally, we introduce the *thermal activation function* $\xi(\cdot)$, which models the aforementioned decay of sinusoidal components (implied by $\mathbf{W}_1$) over time:

$$\xi(\mathbf{z}, \nu, \kappa, t) = \sin(\mathbf{z}) \cdot \exp(-|\nu|^2 \kappa t). \tag{2}$$

Here, $|\nu| = |\nu(\mathbf{W}_1)|$ denotes the row-wise Euclidean norm of $\mathbf{W}_1$, representing the magnitudes of the implied wave numbers (frequencies). Interestingly, Equation 1 constitutes the solution of the isotropic heat equation $\frac{\partial \Phi}{\partial t} = \kappa \cdot \nabla_{\mathbf{x}}^2 \Phi$, as derived in Appendix A. We therefore refer to this MLP as a *neural heat field*.

There is an ideal bijection between the desired sampling rate $f_s$ and $t$. At $t = 0$ no filtering takes place, implying a continuous signal ($f_s \to \infty$). A low-pass filtered version of the signal is observed for $t > 0$. To obtain a desired level of anti-aliasing, we only need to compute the corresponding value of $t$. The

relationship between the cut-off frequency of the filter and $t$ is controlled by a global diffusivity constant $\kappa$, which defines how fast components of different frequencies decay over time in the underlying PDE model. We can freely set $\kappa$ to any positive number, but for simplicity, and without loss of generality, we set $\kappa$ in our theoretical derivations such that the native resolution of the (observed, discrete) signal $\mathcal{D}$ corresponds to $t = 1$. Assuming equal sampling rate $f_s$ along both coordinate axes, the optimal value of $\kappa$ then evaluates to

$$\kappa = \frac{\ln(4)}{2 f_s^2 \pi^2}. \tag{3}$$

To subsample the signal $\mathcal{D}$ by a factor $S$, the field $\Phi$ should be sampled at

$$t = S^2, \tag{4}$$

where $S$ is the subsampling rate, *i.e.*, the inverse of the scaling factor. In other words, the equation above defines the correct value for $t$ according to the scaling factor at which the field should be sampled. For a derivation of these values and a demonstration of the filtering mechanism of neural heat fields, see Section A.

## 3.2 Learning a Super-Resolution Prior

The multi-scale signal representation inherent in neural heat fields is a natural match for ASR. Still, two challenges must be addressed. First, our formulation restricts the choice of architecture to MLPs with a single hidden layer. Second, while it theoretically guarantees the downsampling operation ($t > 1$), the upsampling operation ($0 < t < 1$) remains ill-posed. To narrow down the (infinite) solution space to a unique result, a prior must either be defined in an unsupervised fashion or learned from data. Our solution to both challenges is to condition local fields with a hypernetwork $\Psi : \mathbb{R}^{W \times H \times C} \to \mathbb{R}^{W \times H \times N}$. First, a standard backbone, as used in previous work (Chen et al., 2021; Lee & Jin, 2022; Vasconcelos et al., 2023; Cao et al., 2023; Chen et al., 2023), extracts image features from the low-resolution input image. Then, the hypernetwork maps these features to the $N$ parameters of each local neural heat field. As originally proposed by LIIF (Chen et al., 2021) and adopted by recent ASR methods (Lee & Jin, 2022; Cao et al., 2023; Vasconcelos et al., 2023; Chen et al., 2023), each local field spans the area of one pixel of the low-resolution input. It is important that even though the fields themselves model only a local part of the image, the hypernetwork informs them with contextual features collected over a large receptive field.

During training, the local fields are supervised with values of high-resolution target pixels at the appropriate spatial coordinates $\mathbf{x}$ and time index $t$ (ensuring that the signal is correctly blurred for the target resolution), and the entire architecture is optimized end-to-end. In practice, we directly optimize a single global frequency bank $\mathbf{W}_1$, rather than having the hypernetwork predict a separate $\mathbf{W}_1$ for each low-resolution pixel. Not only does this better fit the idea to represent the signal with a single, consistent basis, it also reduces the total parameter count.

The described scheme, which we call *Thera*, is depicted in Figure 3. It allows for arbitrary-scale super-resolution, combining the multi-scale signal representation within neural heat fields with the expressivity of proven feature extraction backbones for SR and image restoration. As the entire network is trained end-to-end, the feature extractor can learn super-resolution priors for a whole range of resolutions covered by the training data. *E.g.*, a network trained with scaling factors up to $\times 4$ will encode priors that enable us to observe the field at $t > \frac{1}{16}$. By training on multiple resolutions, we can also make $\kappa$ a trainable parameter that allows the network to adapt to different downsampling operators. Finally, we set the bias terms for the three color channels of every local field $\Phi$ (*i.e.*, $\mathbf{b}_2$) to the RGB values of the associated low-resolution pixel. Thus, the hypernetwork only predicts field-wise phase shifts $\mathbf{b}_1$ and amplitudes $\mathbf{W}_2$.

## 3.3 Total Variation at $t = 0$

To allow *Thera* to better generalize to higher, out-of-domain scaling factors, we can place an unsupervised regularizer at $t = 0$. Note that this is a *prior on the continuous signal itself* – something that, to the best of our knowledge, sets *Thera* apart from all previous methods. In our implementation it takes the form of a total variation (TV) loss term, well known to promote piece-wise constant signals that describe natural

images well (Chugunov et al., 2024). We use an $\ell^1$ variant of TV,

$$\mathcal{L}_{\text{TV}}(\Phi(\mathbf{x}, 0)) = \mathbb{E}_{\mathbf{x}}[|\nabla\Phi(\mathbf{x}, 0)|]. \tag{5}$$

Given our continuous signal representation, $\nabla\Phi(\mathbf{x}, 0)$ can be computed analytically by automatic differentiation, rather than falling back to a neighborhood approximation as in most previous work (Rudin et al., 1992). We further motivate this approach in Figure 5, which demonstrates that our method faithfully recovers the gradients of super-resolved images.

### 3.4 Implementation and Training

*Thera* is implemented in JAX (Bradbury et al., 2018). Similar to prior work (Chen et al., 2021; Lee & Jin, 2022; Cao et al., 2023; Zhu et al., 2025), we randomly sample a scaling factor $r \sim \mathcal{U}(1.2, 4)$ for each image during training, then randomly crop an area of size $(48r)^2$ pixels as the target patch, from which the source is generated by bicubic downsampling to size $48^2$. As corresponding targets, $48^2$ random pixels are sampled from the target patch. We train with standard augmentations (random flipping, rotation, and resizing), using the Adam optimizer (Kingma & Ba, 2015) with a batch size of 16 for $5 \times 10^6$ iterations, with initial learning rate $10^{-4}$, $\beta_1 = 0.9$, $\beta_2 = 0.999$ and $\epsilon = 10^{-8}$. The learning rate is decayed to zero according to a cosine annealing schedule (Loshchilov & Hutter, 2016). We use MAE as reconstruction loss, to which the TV loss from Eq. 5 is added with a weight of $10^{-4}$. Like previous work (Timofte et al., 2016; Lim et al., 2017; Vasconcelos et al., 2023), we employ geometric self-ensembling (GSE) instead of the local self-ensembling introduced in LIIF (Chen et al., 2021). In GSE, the results for four rotated versions of the input are averaged at test time. Including reflections did not improve performance.

## 4 Results

Throughout this section we evaluate three variants of our method, which differ solely in the size of the hypernetwork and the number of field parameters:

- *Thera Air*: A tiny version with the number of globally shared components in $\mathbf{W}_1$ set to 32, and the hypernetwork being a single $1 \times 1$ convolution that maps features to field parameters. This version adds only 8,256 parameters on top of the backbone.

- *Thera Plus*: A balanced version that employs an efficient ConvNeXt-based (Liu et al., 2022) hypernetwork. Its parameter count of $\approx$1.41 M matches that of recent medium-sized competitors like Cao et al. (2023).

- *Thera Pro*: The strongest version uses a high-capacity, attention-based hypernetwork. Its added parameter count is $\approx$4.63 M, still less than the most recent competitor (Zhu et al., 2025) and much smaller than Chen et al. (2023), both attention-based.

**Datasets and metrics.** Following previous work, our models are trained with the DIV2K (Agustsson & Timofte, 2017) training set, consisting of 800 high-resolution RGB images of diverse scenes. We report evaluation metrics on the official DIV2K validation split as well as on standard benchmark datasets: Set5 (Bevilacqua et al., 2012), Set14 (Zeyde et al., 2012), BSDS100 (Martin et al., 2001), Urban100 (Huang et al., 2015), and Manga109 (Matsui et al., 2017). Following prior work, we use peak signal-to-noise ratio (PSNR, in decibels) as the main evaluation metric and compute it in RGB space for DIV2K and on the luminance (Y) channel of the YCbCr representation for benchmark datasets. Additional quantitative results are given in Appendix C. Not all numbers could be computed for competing methods for which code or checkpoints were not publicly shared (see Appendix H).

**Backbones.** We combine each of the three variants of our method with two standard backbones for super-resolution and image restoration, as done in previous work: *(i)* EDSR-baseline (Lim et al., 2017) (1.22 M parameters) and *(ii)* RDN (Zhang et al., 2018) (22.0 M parameters).

Table 1: Quantitative comparison of peak signal-to-noise ratio (PSNR, in dB) obtained by various methods on the held-out DIV2K validation set. The highest PSNR value per backbone and scaling factor is **bold** and the second highest is underlined.

| Backbone (params.) | Method | Num. of add. params. | In-distribution | | | Out-of-distribution | | | | |
|---|---|---|---|---|---|---|---|---|---|---|
| | | | ×2 | ×3 | ×4 | ×6 | ×12 | ×18 | ×24 | ×30 |
| — | *Bicubic* | — | 31.01 | 28.22 | 26.66 | 24.82 | 22.27 | 21.00 | 20.19 | 19.59 |
| EDSR-baseline (1.22 M) | *+ Sinc interpol.* | — | 34.52 | 30.89 | 28.98 | 26.75 | 23.76 | 22.25 | 21.29 | 20.62 |
| | MetaSR | 0.45 M | 34.64 | 30.93 | 28.92 | 26.61 | 23.55 | 22.03 | 21.06 | 20.37 |
| | LIIF | 0.35 M | 34.67 | 30.96 | 29.00 | 26.75 | 23.71 | 22.17 | 21.18 | 20.48 |
| | LTE | 0.49 M | 34.72 | 31.02 | 29.04 | 26.81 | 23.78 | 22.23 | 21.24 | 20.53 |
| | CUF | 0.30 M | 34.79 | 31.07 | 29.09 | 26.82 | 23.78 | 22.24 | — | — |
| | CiaoSR | 1.43 M | 34.88 | 31.12 | 29.19 | 26.92 | 23.85 | 22.30 | 21.29 | 20.44 |
| | CLIT | 15.7 M | 34.81 | 31.12 | 29.15 | 26.92 | 23.83 | 22.29 | 21.26 | 20.53 |
| | SRNO | 0.80 M | 34.85 | 31.11 | 29.16 | 26.90 | 23.84 | 22.29 | 21.27 | 20.56 |
| | MSIT | 4.83 M | 34.95 | 31.23 | 29.22 | 26.94 | 23.83 | 22.27 | 21.26 | 20.54 |
| | *Thera Air* (ours) | .008 M | 34.75 | 31.09 | 29.10 | 26.84 | 23.80 | 22.26 | 21.26 | 20.56 |
| | *Thera Plus* (ours) | 1.41 M | 34.89 | 31.22 | 29.24 | 26.96 | 23.89 | 22.34 | 21.32 | 20.61 |
| | *Thera Pro* (ours) | 4.63 M | **35.19** | **31.50** | **29.51** | **27.19** | **24.09** | **22.51** | **21.48** | **20.73** |
| RDN (22.0 M) | *+ Sinc interpol.* | — | 34.59 | 31.03 | 29.12 | 26.89 | 23.87 | 22.34 | 21.36 | 20.68 |
| | MetaSR | 0.45 M | 35.00 | 31.27 | 29.25 | 26.88 | 23.73 | 22.18 | 21.17 | 20.47 |
| | LIIF | 0.35 M | 34.99 | 31.26 | 29.27 | 26.99 | 23.89 | 22.34 | 21.31 | 20.59 |
| | LTE | 0.49 M | 35.04 | 31.32 | 29.33 | 27.04 | 23.95 | 22.40 | 21.36 | 20.64 |
| | CUF | 0.30 M | 35.11 | 31.39 | 29.39 | 27.09 | 23.99 | 22.42 | — | — |
| | CiaoSR | 1.43 M | 35.13 | 31.39 | 29.43 | 27.13 | 24.03 | 22.45 | 21.41 | 20.55 |
| | CLIT | 15.7 M | 35.10 | 31.39 | 29.39 | 27.12 | 24.01 | 22.45 | 31.38 | 20.64 |
| | SRNO | 0.80 M | 35.16 | 31.42 | 29.42 | 27.12 | 24.03 | 22.46 | 21.41 | 20.68 |
| | MSIT | 4.83 M | 35.16 | 31.42 | 29.42 | 27.11 | 23.99 | 22.42 | 21.37 | 20.65 |
| | *Thera Air* (ours) | .008 M | 35.06 | 31.42 | 29.43 | 27.13 | 24.04 | 22.48 | 21.44 | 20.71 |
| | *Thera Plus* (ours) | 1.41 M | 35.00 | 31.40 | 29.44 | 27.16 | 24.06 | 22.49 | 21.45 | 20.71 |
| | *Thera Pro* (ours) | 4.63 M | **35.25** | **31.56** | **29.57** | **27.25** | **24.14** | **22.56** | **21.52** | **20.77** |

Table 2: Results on common benchmark datasets for in-distribution scale factors with an RDN (Zhang et al., 2018) backbone. The numbers represent PSNR in dB, calculated on the luminance (Y) channel of the YCbCr representation following previous work.

| Method | Set5 | | | Set14 | | | B100 | | | Urban100 | | | Manga109 | | |
|---|---|---|---|---|---|---|---|---|---|---|---|---|---|---|---|
| | ×2 | ×3 | ×4 | ×2 | ×3 | ×4 | ×2 | ×3 | ×4 | ×2 | ×3 | ×4 | ×2 | ×3 | ×4 |
| MetaSR | 38.22 | 34.63 | 32.38 | 33.98 | 30.54 | 28.78 | 32.33 | 29.26 | 27.71 | 32.92 | 28.82 | 26.55 | — | — | — |
| LIIF | 38.17 | 34.68 | 32.50 | 33.97 | 30.53 | 28.80 | 32.32 | 29.26 | 27.74 | 32.87 | 28.82 | 26.68 | 39.26 | 34.21 | 31.20 |
| LTE | 38.23 | 34.72 | 32.61 | 34.09 | 30.58 | 28.88 | 32.36 | 29.30 | 27.77 | 33.04 | 28.97 | 26.81 | 39.28 | 34.32 | 31.30 |
| CUF | 38.28 | 34.80 | 32.63 | 34.08 | 30.65 | 28.92 | 32.39 | 29.33 | 27.80 | 33.16 | 29.05 | 26.87 | — | — | — |
| CiaoSR | 38.29 | 34.85 | 32.66 | 34.22 | 30.65 | 28.93 | 32.41 | 29.34 | 27.83 | 33.30 | 29.17 | 27.11 | 39.51 | 34.57 | 31.57 |
| CLIT | 38.26 | 34.80 | 32.69 | 34.21 | 30.66 | 28.98 | 32.39 | 29.34 | 27.82 | 33.13 | 29.04 | 26.91 | — | — | — |
| SRNO | 38.32 | 34.84 | 32.69 | 34.27 | 30.71 | 28.97 | 32.43 | 29.37 | 27.83 | 33.33 | 29.14 | 26.98 | 39.52 | 34.67 | 31.61 |
| MSIT | 38.31 | 34.85 | 32.72 | 34.26 | 30.70 | 28.97 | 32.42 | 29.35 | 27.81 | 33.27 | 29.14 | 26.93 | 39.44 | 34.62 | 31.58 |
| *Thera Air* | 38.18 | 34.75 | 32.60 | 34.13 | 30.70 | 28.95 | 32.33 | 29.29 | 27.80 | 33.16 | 29.14 | 26.91 | 39.03 | 34.57 | 31.66 |
| *Thera Plus* | 38.11 | 34.67 | 32.56 | 34.20 | 30.67 | 28.97 | 32.26 | 29.28 | 27.81 | 33.14 | 29.15 | 26.97 | 38.69 | 34.42 | 31.65 |
| *Thera Pro* | **38.36** | **34.88** | **32.79** | **34.43** | **30.85** | **29.08** | **32.46** | **29.39** | **27.87** | **33.63** | **29.58** | **27.26** | **39.62** | **34.98** | **31.98** |

## 4.1 Super-Resolution Performance

**Quantitative results.** We first evaluate the three variants of our method on the held-out DIV2K validation set, following the setup described above. Table 1 shows PSNR values for all tested methods, for both in-distribution (×2 to ×4) and out-of-distribution (×6 to ×30) scaling factors. *Thera Pro* outperforms all competing methods at all scaling factors, often by a substantial margin (*e.g.*, 29.51 *vs.* 29.22 on EDSR ×4), even though its parameter overhead on top of the backbone is lower compared to the second-best method MSIT (Zhu et al., 2025), and less than a third of CLIT (Chen et al., 2023). Interestingly, even our minimal

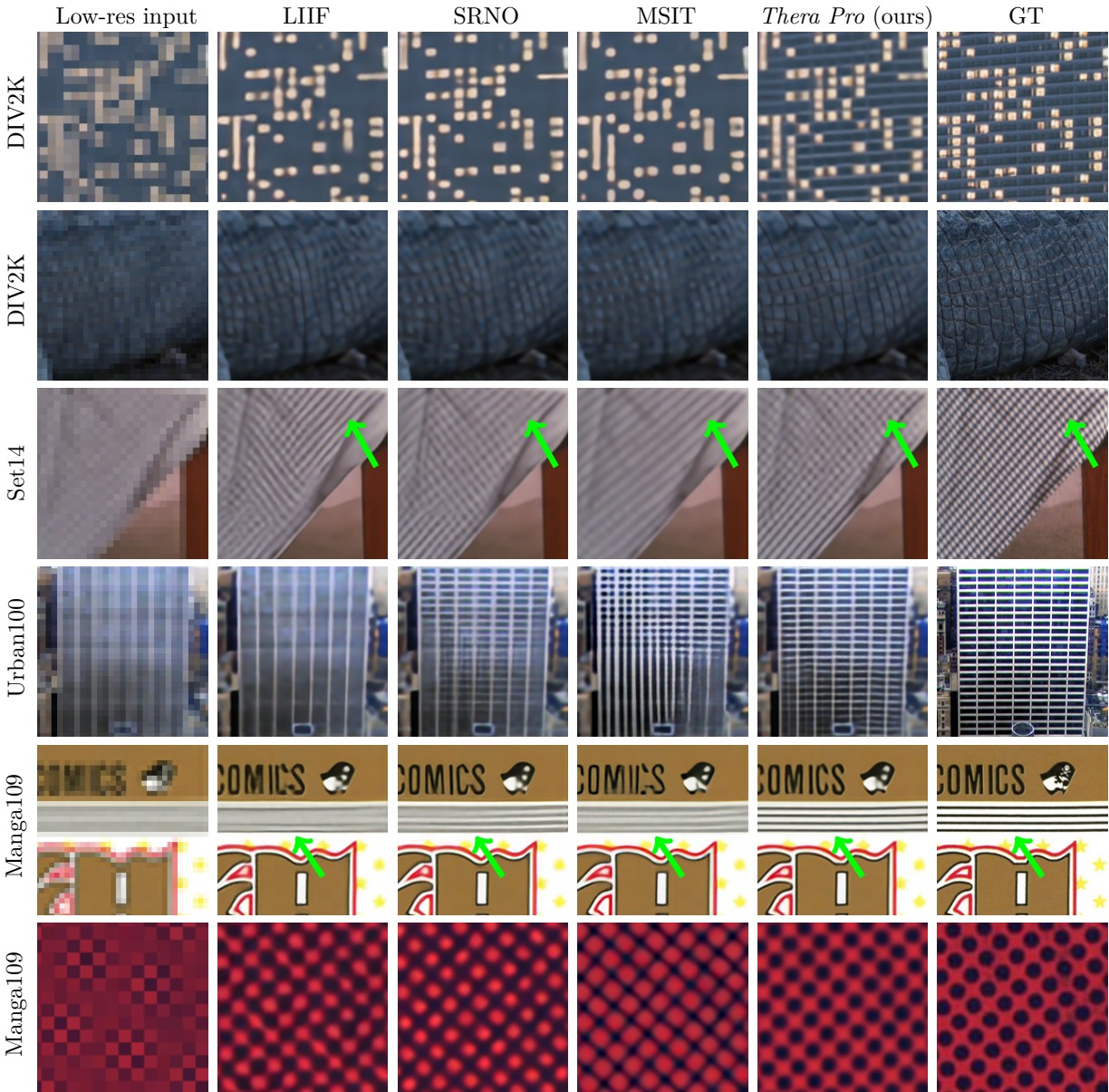

Figure 4: Qualitative examples for a representative ×6 scale factor, with an RDN (Zhang et al., 2018) backbone for all methods. Best viewed zoomed in.

variant *Thera Air* – with only about 8000 parameters on top of the backbone – performs on par with or better than methods of much higher parameter count. This supports our claim that hard-wiring a theoretically principled sampling model, which rules out signal aliasing, enables better generalization and higher-fidelity reconstruction. For comparison with conventional interpolation, we also report numbers obtained with Lanczos (sinc) resampling on top of the respective ×4 backbone. This baseline is consistently outperformed by dedicated ASR methods, indicating that the latter do learn scale-specific priors.

Like earlier work, we further report the performance of *Thera* on five popular benchmark datasets with an RDN backbone in Table 2. Our method again outperforms all competing methods in all settings, often substantially (*e.g.*, 29.58 *vs.* 29.14 on Urban100 ×3). We hypothesize that *Thera*'s hard-wired PSF is also beneficial when generalizing to unseen datasets. Once again we observe that the performance of *Thera Air* is often comparable to that of methods with orders of magnitude higher parameter counts.

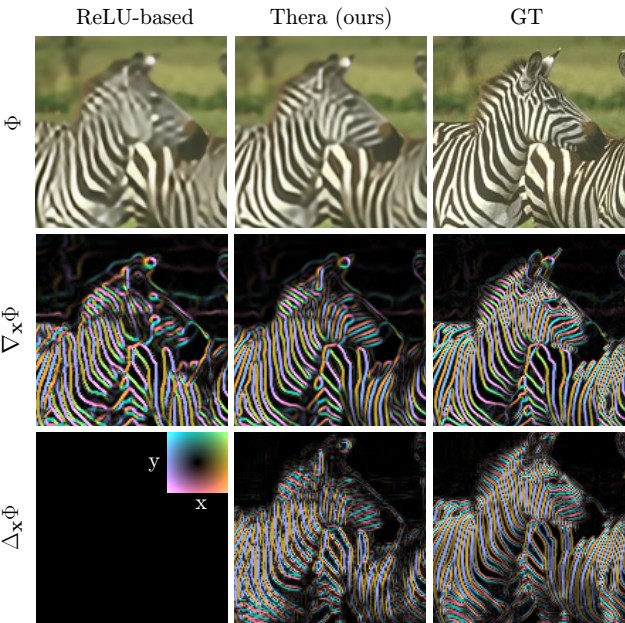

Figure 5: *Thera* reconstructs a signal $\Phi$ and its gradient $\nabla_{\mathbf{x}}\Phi$ more faithfully than a ReLU-based competitor (Chen et al., 2021). Due to its natural, Fourier-inspired representation, *Thera* is also infinitely differentiable, while ReLU-based competitors approximate the signal as a piecewise-linear function with null higher derivatives (last row).

**Qualitative results.** Upon visual inspection – see Figure 4 for examples – we observe that *Thera* produces results that are both perceptually convincing and more correct, particularly in the presence of repeating structures. Neural heat fields enable *Thera* to reproduce a high level of detail without suffering from aliasing, no matter the sampling scale (see also Figure 9 in Appendix B).

**Fidelity of the Signal and its Derivatives.** Neural fields with periodic activation functions have been shown to be superior when it comes to fitting high-resolution, natural signals, and to correctly recovering their derivatives (Sitzmann et al., 2020b). We observe similar effects for *Thera*, whose thermal activations at $t = 0$ can be seen as a special case of periodic activations, *cf*. Figure 5. In fact, due to the use of thermal activations – and unlike all prior work based on multi-layer ReLU-activated fields – *Thera* is infinitely differentiable.

### 4.2 Ablation Studies

In Table 3, we ablate individual components and design choices of our method to understand their contributions to overall performance. The comparisons use *Thera Plus* with EDSR backbone, and are representative of all variants.

**Single scale training.** We run three experiments using a single scale ($\times 2$, $\times 3$, $\times 4$) to test how this affects scale generalization. $\kappa$ was fixed at the theoretically derived value for these experiments, as multi-scale training is required to optimize it. As expected, we observe equal or even superior performance of single-scale training when tested at the training scale (marked in yellow in Table 3), but a significant drop compared to the default multi-scale version when generalizing to other scaling factors.

**Trainable $\kappa$.** Fixing $\kappa$ at the theoretically derived value (Equation 3) leads to a small drop in performance. This suggests that there remain effects that are not accounted for by our proposed observation model, albeit very minor.

Table 3: Ablation study using *Thera Plus* (w/ EDSR-baseline)

| Experiment | In-distribution | | | Out-of-distribution | | |
|---|---|---|---|---|---|---|
| | ×2 | ×3 | ×4 | ×6 | ×18 | ×30 |
| *Thera Plus* | 34.89 | 31.22 | 29.24 | 26.96 | 22.34 | 20.61 |
| ×2 only, fixed $\kappa$ | 35.00 | 30.96 | 28.76 | 26.34 | 21.87 | 20.25 |
| ×3 only, fixed $\kappa$ | 34.70 | 31.25 | 29.25 | 26.90 | 22.23 | 20.53 |
| ×4 only, fixed $\kappa$ | 34.36 | 31.18 | 29.25 | 26.98 | 22.31 | 20.58 |
| Fixed $\kappa$ | 34.85 | 31.22 | 29.24 | 26.95 | 22.29 | 20.56 |
| No GSE | 34.81 | 31.15 | 29.18 | 26.91 | 22.29 | 20.57 |
| No TV prior | 34.89 | 31.23 | 29.24 | 26.87 | 20.42 | 18.68 |
| ReLU instead $\xi$ | 34.80 | 31.11 | 29.13 | 26.87 | 22.28 | 20.57 |
| Predicted comps. | 34.90 | 31.23 | 29.25 | 26.97 | 22.34 | 20.61 |

**Geometric self-ensemble.** In line with previous work (Timofte et al., 2016; Lim et al., 2017; Vasconcelos et al., 2023) we see a notable performance boost with geometric self-ensembling. Note, though, if an application prioritizes inference speed over quality this add-on can be disabled at test-time without re-training the network.

**Total variation prior.** The regularizer has a negligible effect for in-domain scaling factors, but performance degrades significantly without it for out-of-distribution scales.

**Thermal activations.** We replace thermal activations (Equation 2) with standard ReLU activations. What remains is only the hypernetwork controlling the parameters of the local fields. A consistent loss in performance shows the impact of the proposed thermal activation underlying our multi-scale representation.

**Shared components.** Predicting $W_1$ along with $b_1$ and $W_2$ leads to negligible gains. This comes at the cost of doubling the amount of field parameters that the hypernetwork predicts. Thus, *Thera* uses a shared, global frequency bank.

### 4.3 Limitations and Future Work

Neural heat fields as introduced in this paper, and by extension *Thera*, come with relatively strict architectural requirements that currently only allow for a single hidden layer in the neural field. While this can be beneficial from a computational standpoint, it limits hierarchical feature learning and potentially makes modeling of complex non-linear relations harder than necessary. Nonetheless, as our experiments show, the current neural heat field architecture does easily have enough capacity to model local, subpixel information for the scaling factor range discussed in this paper. We have compensated for the relatively less expressive fields with a higher-capacity hypernetwork, and we speculate that there may be ways to extend the signal-theoretic guarantees of *Thera* to multi-layer architectures in future work. This could result in even higher parameter efficiency, and potentially better generalization. We also expect that more advanced priors than TV could be even more effective at regularizing $\Phi$. Priors at $t = 0$, made possible by *Thera*, have the potential to regularize the *continuous signal itself*, and therefore improve SR quality for all scaling factors.

## 5 Conclusion

We have developed a novel paradigm for arbitrary-scale super-resolution by combining traditional signals theory with modern implicit neural representations. Our proposed neural heat fields implicitly describe an image as a combination of sinusoidal components, which can be selectively modulated according to their frequency to perform Gaussian filtering (anti-aliasing) between scales analytically, with negligible overhead. Our experimental evaluation shows that *Thera*, our ASR method based on neural heat fields, consistently outperforms competing methods. At the same time, it is more parameter-efficient and offers theoretical

guarantees w.r.t. aliasing. We believe that *Thera*-style representations could benefit other computer vision tasks and hope to inspire further research into neural methods that integrate physically meaningful and theoretically grounded observation models.

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

# A Theory

## A.1 Preliminaries

As was described in Section 3.1, the idea underlying our neural heat field with thermal activations is to formulate a neural field $\Phi(\mathbf{x}, t)$, with $\mathbf{x}$ being the 2-dimensional spatial coordinates $(x_1, x_2)$, such that $\Phi$ follows the heat equation:

$$\frac{\partial \Phi}{\partial t} = \kappa \cdot \nabla_{\mathbf{x}}^2 \Phi = \kappa \cdot \left( \frac{\partial^2 \Phi}{\partial x_1^2} + \frac{\partial^2 \Phi}{\partial x_2^2} \right) \ . \tag{6}$$

The reason for this is that the analytical solution to the (isotropic) heat equation can be modeled as a convolution of the initial state $\Phi(\mathbf{x}, 0)$ with a Gaussian kernel

$$g(\mathbf{x}, t) = \frac{1}{4\pi\kappa t} \cdot \exp\left(-\frac{x_1^2 + x_2^2}{4\kappa t}\right). \tag{7}$$

By fitting the data (image $I$) at $\Phi(\mathbf{x}, 1)$, we are assuming a Gaussian point spread function (PSF) with the shape

$$\text{PSF}(\mathbf{x}) = \frac{1}{4\pi\kappa} \cdot \exp\left(-\frac{x_1^2 + x_2^2}{4\kappa}\right). \tag{8}$$

In this formulation, we attempt to recover a "pure" signal at $t = 0$ or higher sampling rates $0 < t < 1$ given an observation at $t = 1$. Note that

$$\Phi(\mathbf{x}, t) \text{ is } \begin{cases} \text{meaningless,} & \text{if } t < 0 \\ \text{pure signal,} & \text{if } t = 0 \\ \text{ill-posed,} & \text{if } 0 < t < 1 \\ I, & \text{if } t = 1 \\ \text{well-posed,} & \text{if } t > 1 \end{cases} \tag{9}$$

The ill-posed problem for $0 < t < 1$ is the interesting case, where this formulation relates to super-resolution. The super-resolution algorithm should somehow condition the solution space to find the appropriate solution in this domain.

For all the formulations here, we define the image $I$ to correspond to the coordinates $x_1, x_2 \in [-0.5, 0.5]$.

## A.2 Thermal Diffusivity Coefficient

To use the above formulations, we need to compute the thermal diffusivity coefficient $\kappa$. One way to do so is to match the cut-off frequency of the filter in Equation 7 at $t = 1$ to the well-known Nyquist frequency given by the image's sampling rate. We take the cut-off frequency of the Gaussian filter defined in Equation 7 to be the frequency whose amplitude is halved, which is

$$f_c = \sqrt{\ln(4)} \cdot \sigma_f = \frac{\sqrt{\ln(4)}}{2\pi\sqrt{2kt}} \tag{10}$$

For the signal compressed into the domain $[-0.5, 0.5]$, we can compute the Nyquist frequency to be

$$f_{\text{Nyquist}} = \frac{N}{2}, \tag{11}$$

where $N$ is the number of samples along a given dimension. This formulation assumes even sampling over $x_1$ and $x_2$. To extend this formulation to non-square images, it would be necessary to change the shape of the signal's domain in order to maintain even sampling in all spatial dimensions.

If we solve for $f_c = f_{\text{Nyquist}}$ at $t = 1$, we get

$$\kappa = \frac{\ln(4)}{2\pi^2 N^2}. \tag{12}$$

For our proposed *Thera* formulation, we want $\Phi$ to contain a single pixel at $t = 1$. This is the pixel from the low-resolution input which will become $SR^2$ pixels for super-resolution with a scaling factor of $SR$. Therefore we initialize $\kappa$ with

$$\kappa = \frac{\ln(4)}{2\pi^2}. \tag{13}$$

Note that the exact value of $\kappa$ will depend on the characteristics of the system that is being modeled and the anti-aliasing filter that was used (or is assumed). Lower values of $\kappa$ allow for sharper signals to be represented at any given value of $t$, but are also more prone to aliasing.

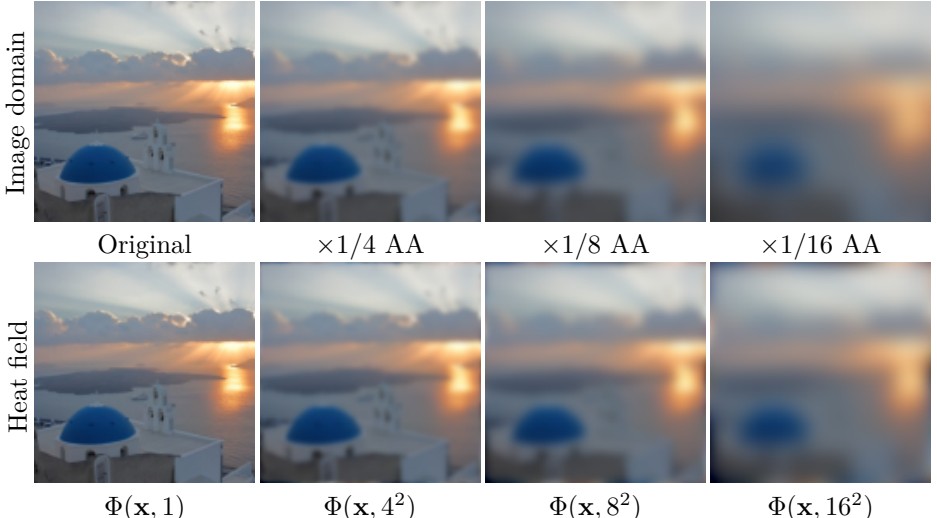

Figure 6: Comparison between a theoretical anti-aliasing filter (top row) and anti-aliasing with neural heat fields (bottom row), which is computed without convolutions or over-sampling. The heat field was supervised at $\Phi(\mathbf{x}, 1)$ *only.*

Finally, we would like to highlight that Equation 10 is specific to the case where $\mathbf{x}$ is 2-dimensional. The theoretically ideal value of $\kappa$ is the only part of our formulation that does not directly apply when using our field's formulation in spaces with numbers of spatial dimensions other than 2. Nonetheless, computing $\kappa$ for other cases would be a simple matter of repeating the steps above with the formulas for a Gaussian filter with the appropriate number of dimensions.

### A.3 Relationship Between t and s

Assuming that the field is learned appropriately, we still need to know at what time $t$ we should sample from to obtain the correct (aliasing-free) signal for a different sampling rate. If we define $S$ to be the **subsampling** rate (*i.e.*, if our base image has $N = 128$ and we want to subsample it down to $N = 64$, we have $S = 2$) we need to find $t$ such that $f_{\text{Nyquist}}$ scales by $1/S$. Using Equation 10 and Equation 11, we can easily find the quadratic relationship

$$t = S^2. \tag{14}$$

For instance, if we want to **upsample** the image by a factor of 2, we should use $t = 0.5^2 = 0.25$. Thus, $0 < t < 1$ refers to super-resolution, while $t > 1$ refers to downsampling. This is intuitive: As $t$ grows, the image becomes blurrier (the Gaussian kernel gets wider), which corresponds to stronger low-pass filters and therefore lower sampling rates.

Figure 6 shows an example where we fit a neural heat field at $t = 1$ to the image. After training, any low-pass filtered version of the image can be generated by setting $t$ according to Equation 4. We emphasize that: *(i)* Computing these filtered images requires no over-sampling or convolutions; *(ii)* The computational cost does **not** depend on the size of the blur kernel or on $t$; *(iii)* Given $\Phi(\mathbf{x}, t_0)$, the filtered versions $\Phi(\mathbf{x}, t)$ are known for any $t \geq t_0$.

In Figure 7, we show four local neural heat fields in which aliasing would occur without the anti-aliasing mechanism modeled by thermal activations. Note that aliasing is not always as obvious to the eye as Moiré patterns: in the cases shown in the figure, it would simply mean that sampling the center location at $t = 0$ would not be representative of the pixel's footprint. Figure 8 further illustrates how such blur is equivalent to a scale-appropriate anti-aliasing filter.

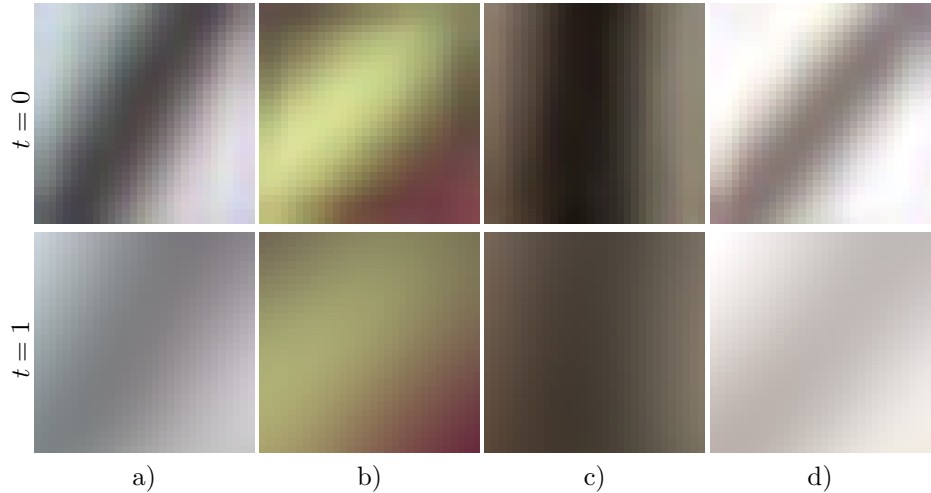

Figure 7: Example situations where aliasing would occur without the suppression of high frequencies as modeled by neural heat fields. Sampling the center location at $t = 0$ would not be representative of the pixel's footprint.

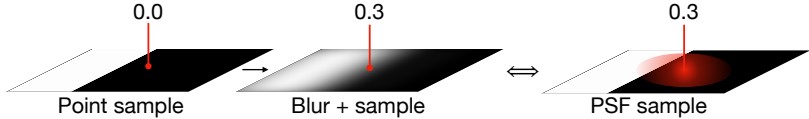

Figure 8: Blurring an image before a point-wise sample is equivalent to observing with a PSF equivalent to the blur kernel.

### A.4 Other Filters

In theory, the formulation presented in Section 3.1 allows us to use any low-pass filter we want, since we can modulate different components freely. Gaussian filters are an obvious choice since they are often used as anti-aliasing filters, and since they are fully defined by a single parameter, the standard deviation. Initial explorations of a sharp low-pass filter that completely removes components above $f_{\text{Nyquist}}$ led to a performance reduction, likely due to the associated effect on gradients during training. It remains an open question whether more complex filters (*e.g.*, Butterworth) would improve the current formulation in any way. For quantitative evaluations, this is unlikely, since the downsampling operations use Gaussian anti-aliasing, but in real-world applications or other scenarios, this may be desirable.

### A.5 Initialization of Components

We have noticed during our experiments that the initialization of the components, $\mathbf{W}_1$ in Equation 1, is important. Sitzmann et al. (2020b) made similar observations when periodic activation functions were first used for neural fields. The final distribution of frequencies $|\nu(\mathbf{W}_1)|$ did not change much during training. Thus, we choose to initialize $\mathbf{W}_1$ such that

$$p(|\nu(\mathbf{w}_1)|) \propto |\nu(\mathbf{w}_1)| \tag{15}$$

up to a given maximum frequency, allotting more components to higher frequencies. See code for more details.

## B Continuous Upsampling Example

In Figure 9, we provide a practical showcase of the continuous upsampling capabilities of our method.

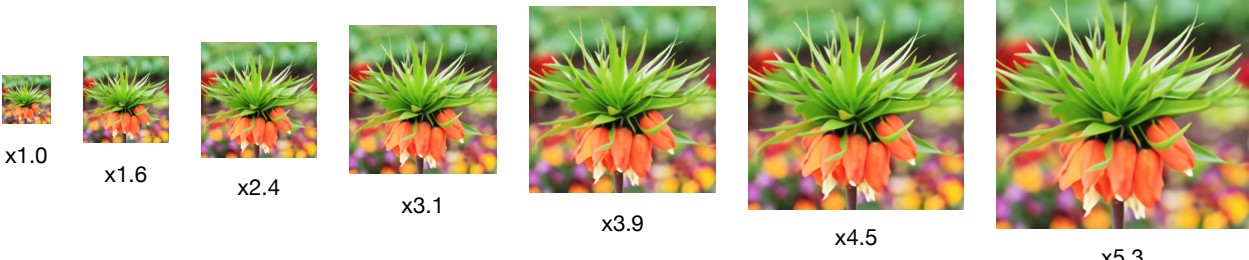

x1.0 x1.6 x2.4 x3.1 x3.9 x4.5 x5.3

Figure 9: Showcase of multiscale upsampling using *Thera Pro* with a RDN (Zhang et al., 2018) backbone, shown with non-integer scaling factors.

# C  Additional Quantitative Results

## C.1  Further Metrics

Table 4 shows SSIM metrics on the DIV2K validation set, which complement the PSNR values reported in Table 1. We use the SSIM implementation from torchmetrics (Detlefsen et al., 2022). We observe strong performance of *Thera*, although overall there is relatively little variance across all methods using this metric.

Table 4: SSIM scores (higher is better) for several methods and scaling factors evaluated on the (hold-out) DIV2K validation set. We use the RDN backbone for all models. Some methods did not provide checkpoints at the time of writing, see Appendix H.

|  | In-distribution | | | OOD | |
|---|---|---|---|---|---|
|  | ×2 | ×3 | ×4 | ×6 | ×8 |
| LIIF | .934 | .866 | .804 | .704 | .636 |
| LTE | .936 | .868 | .807 | .707 | .639 |
| CiaoSR | .938 | .871 | .814 | .718 | .651 |
| SRNO | .937 | .873 | .819 | .738 | .685 |
| MSIT | .942 | .882 | .829 | .751 | .700 |
| *Thera Pro* | **.943** | **.884** | **.833** | **.757** | **.706** |
| *Thera Pro* (std.) | ±2.0e-5 | ±4.0e-5 | ±8.9e-5 | ±9.7e-5 | ±5.5e-5 |

Table 5: Evaluation of GMSD (lower is better) on DIV2K for various methods.

| Backbone | Model | In-distribution | | | Out-of-distribution | | | | |
|---|---|---|---|---|---|---|---|---|---|
|  |  | ×2 | ×3 | ×4 | ×6 | ×12 | ×18 | ×24 | ×30 |
| EDSR-b. | LIIF | 0.0065 | 0.0354 | 0.0667 | 0.1192 | 0.1976 | 0.2319 | 0.2516 | 0.2645 |
|  | LTE | 0.0064 | 0.0351 | 0.0664 | 0.1187 | 0.1970 | 0.2316 | 0.2515 | 0.2645 |
|  | Ciao | 0.0063 | 0.0340 | 0.0641 | 0.1159 | 0.1942 | 0.2295 | 0.2497 | 0.2635 |
|  | SRNO | 0.0063 | 0.0344 | 0.0648 | 0.1165 | 0.1947 | 0.2296 | 0.2499 | 0.2631 |
|  | MSIT | 0.0061 | 0.0334 | 0.0636 | 0.1156 | 0.1944 | 0.2298 | 0.2502 | 0.2634 |
|  | *Thera Pro* | **0.0057** | **0.0314** | **0.0596** | **0.1094** | **0.1878** | **0.2242** | **0.2451** | **0.2591** |
|  | *Thera Pro* (std.) | ±1.0e-5 | ±4.1e-5 | ±7.6e-5 | ±1.6e-5 | ±7.1e-5 | ±5.9e-5 | ±1.2e-4 | ±1.6e-4 |
| RDN | LIIF | 0.0060 | 0.0330 | 0.0628 | 0.1145 | 0.1934 | 0.2285 | 0.2489 | 0.2622 |
|  | LTE | 0.0058 | 0.0327 | 0.0622 | 0.1135 | 0.1925 | 0.2280 | 0.2487 | 0.2621 |
|  | Ciao | 0.0058 | 0.0321 | 0.0609 | 0.1114 | 0.1897 | 0.2258 | 0.2470 | 0.2614 |
|  | SRNO | 0.0057 | 0.0321 | 0.0611 | 0.1118 | 0.1906 | 0.2266 | 0.2477 | 0.2613 |
|  | MSIT | 0.0057 | 0.0321 | 0.0610 | 0.1122 | 0.1920 | 0.2288 | 0.2501 | 0.2641 |
|  | *Thera Pro* | **0.0055** | **0.0309** | **0.0588** | **0.1083** | **0.1869** | **0.2235** | **0.2446** | **0.2588** |
|  | *Thera Pro* (std.) | ±7.9e-6 | ±1.4e-5 | ±4.0e-5 | ±3.1e-5 | ±3.7e-5 | ±2.0e-5 | ±6.3e-5 | ±9.0e-5 |

Table 6: PSNR (Y channel) on common benchmark datasets for out-of-distribution scale factors, with an RDN (Zhang et al., 2018) backbone. For some methods, code and/or checkpoints were not publicly available, see Appendix H.

| Method | Set5 | | Set14 | | B100 | | Urban100 | | Manga109 | |
|---|---|---|---|---|---|---|---|---|---|---|
| | ×6 | ×8 | ×6 | ×8 | ×6 | ×8 | ×6 | ×8 | ×6 | ×8 |
| MetaSR | 29.04 | 29.96 | 26.51 | 24.97 | 25.90 | 24.83 | 23.99 | 22.59 | — | — |
| LIIF | 29.15 | 27.14 | 26.64 | 25.15 | 25.98 | 24.91 | 24.20 | 22.79 | 27.33 | 25.04 |
| LTE | 29.32 | 27.26 | 26.71 | 25.16 | 26.01 | 24.95 | 24.28 | 22.88 | 27.49 | 25.12 |
| CUF | 29.27 | — | 26.74 | — | 26.03 | — | 24.32 | — | — | — |
| CiaoSR | _29.46_ | **27.36** | 26.79 | 25.28 | _26.07_ | _25.00_ | _24.58_ | _23.13_ | 27.70 | 25.40 |
| CLIT | 29.39 | _27.34_ | _26.83_ | _25.35_ | _26.07_ | _25.00_ | 24.43 | 23.03 | — | — |
| SRNO | 29.38 | 27.28 | 26.76 | 25.26 | 26.04 | 24.99 | 24.43 | 23.02 | 27.66 | 25.31 |
| MSIT | 29.34 | 27.29 | 26.75 | 25.26 | 26.05 | 24.98 | 24.43 | 22.99 | 27.61 | 25.26 |
| _Thera Air_ | 29.31 | 27.25 | 26.76 | 25.27 | 26.05 | 24.99 | 24.39 | 23.00 | 27.70 | 25.30 |
| _Thera Plus_ | 29.31 | 27.29 | 26.80 | 25.32 | _26.07_ | _25.00_ | 24.45 | 23.04 | _27.80_ | _25.41_ |
| _Thera Pro_ | **29.51** | _27.34_ | **26.90** | **25.38** | **26.12** | **25.04** | **24.70** | **23.24** | **27.94** | **25.49** |

Table 5 reports Gradient Magnitude Similarity Deviation (Xue et al., 2013) (GMSD) metrics for various methods. _Thera_ reaches significantly lower GMSD for all backbone-scale combinations, indicating more faithful gradient structure. Note that some methods did not provide code/checkpoints and can therefore not be re-evaluated (see Section H).

Furthermore, in Table 6 we show quantitative evaluations which are out of distribution both in terms of data (benchmark datasets) and in terms of scaling factors (above ×4).

## C.2   Parameter Efficiency

In Figure 10 we compare the number of additional parameters and PSNR values for various methods and individual upsampling factors on the DIV2K validation set.

## C.3   Error Bars

Table 7 reports standard deviations for PSNR observed over $N = 3$ training runs that were initialized with different random seeds, complementing the main table. For SSIM and GMSD, we have reported standard deviations alongside the respective metrics in Tables 4 and 5. Standard deviations are in all cases significantly lower than performance improvements over competing methods.

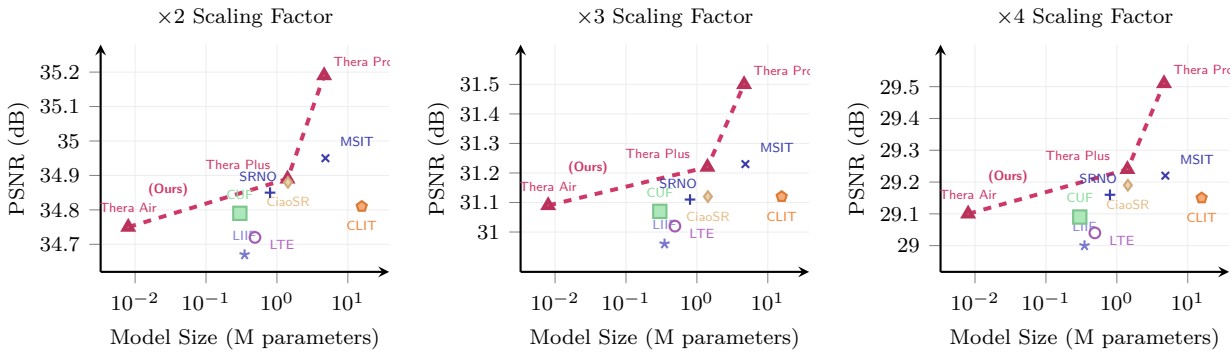

Figure 10: Comparison of ASR methods for different scaling factors (×2, ×3, and ×4) on DIV2K. Thera consistently achieves better performance at lower parameter counts across all scaling factors.

Table 7: Standard deviation of PSNR (in dB) on the DIV2K validation set over $N=3$ runs.

| Backbone | Method | In-distribution | | | Out-of-distribution | | | | |
|---|---|---|---|---|---|---|---|---|---|
| | | ×2 | ×3 | ×4 | ×6 | ×12 | ×18 | ×24 | ×30 |
| EDSR-b. | *Thera Air* | 0.0137 | 0.0143 | 0.0149 | 0.0107 | 0.0084 | 0.0055 | 0.0040 | 0.0040 |
| | *Thera Plus* | 0.0131 | 0.0122 | 0.0114 | 0.0094 | 0.0075 | 0.0049 | 0.0060 | 0.0022 |
| | *Thera Pro* | 0.0063 | 0.0088 | 0.0062 | 0.0034 | 0.0054 | 0.0049 | 0.0056 | 0.0034 |
| RDN | *Thera Air* | 0.0187 | 0.0152 | 0.0113 | 0.0045 | 0.0036 | 0.0054 | 0.0022 | 0.0034 |
| | *Thera Plus* | 0.0045 | 0.0018 | 0.0011 | 0.0011 | 0.0009 | 0.0013 | 0.0026 | 0.0024 |
| | *Thera Pro* | 0.0025 | 0.0009 | 0.0038 | 0.0024 | 0.0021 | 0.0036 | 0.0021 | 0.0012 |

# D  Analysis of Learned Components & Kappa

Figure 11 shows a statistical analysis of the frequency components learned by Thera-Pro with an RDN backbone for the DIV2K training data. Components are distributed uniformly across directions, with fewer components at low frequencies and progressively more components at higher frequencies. This provides more representational capacity for the reconstruction of fine details, analogous to the typical distribution of frequency components for other decompositions such as the Fourier or discrete cosine transforms.

To investigate whether these components form a generalizable basis, we fit heat fields to images from various datasets other than the training data: Urban100, Manga109, and a highly out-of-distribution subset from the fastMRI (Zbontar et al., 2018) medical imaging dataset (single-coil knee validation split, with intensities scaled to [0,1]). In all experiments, we fix the frequency bank to the one shown in Figure 11 and optimize only the scale and shift parameters with the AdamW optimizer for 600 iterations per image, with learning rate 0.001. We use local fields that cover patches of $N \times N$ pixels, with $N \in \{3, 4, 6, 12, 18\}$. Table 8 shows that the pre-trained components reconstruct all datasets with negligible error at smaller local field sizes (in-distribution scales), and still achieve very low errors at larger field sizes up to $18 \times 18$ pixels (out-of-distribution scales).

These numbers are not surprising, as the frequency bank was optimized to fit local fields and can act as an over-parametrized dictionary for smaller field sizes (e.g., $4 \times 4$ GT pixels per field). Notably, it still works relatively well for large OOD sampling factors. Also, there is no obvious difference between the values for natural images and those for MRI images, which suggests that the frequency bank is a general-purpose basis, similar to Fourier or DCT components.

Furthermore, Table 9 reports final, converged values of the thermal diffusivity coefficient $\kappa$. We observe that $\kappa$ is very similar across runs ($\sigma \approx 0.003$), but deviates from the theoretically derived value (for a Gaussian downsampling model, Equation 3) by a factor of $\approx 1/2$. This deviation suggests that the cutoff frequency of the anti-aliasing used in the cubic Mitchell-Netravali filter – used to downsample images during training – does not exactly match the Gaussian one used in the theoretical derivations, being more lenient towards aliasing than the theoretical model (anti-aliasing filters are tuned empirically to balance aliasing against loss of details). The result indicates that Thera is indeed able to tune $\kappa$ as necessary to approximate the downsampling characteristics seen in the training data, in a repeatable manner.

Table 8: Reconstruction MSE when fitting heat field grids of various sizes to out-of-distribution datasets.

| Local field size | 3×3 | 4×4 | 6×6 | 12×12 | 18×18 |
|---|---|---|---|---|---|
| Set5 | 2e-10 | 6e-11 | 1e-11 | 1.37e-6 | 2.16e-4 |
| Urban100 | 4e-9 | 9e-10 | 2e-10 | 2.83e-6 | 7.54e-4 |
| Manga109 | 2e-8 | 4e-9 | 7e-10 | 1.41e-6 | 3.01e-4 |
| fastMRI | 3e-11 | 1e-11 | 1e-11 | 8.83e-6 | 9.23e-4 |

Table 9: Converged values of $\kappa$ for multiple runs.

| Backbone | Model | Converged $\kappa$ |
|---|---|---|
| EDSR-b. | *Thera Air* | 0.0295 |
| | *Thera Plus* | 0.0336 |
| | *Thera Pro* | 0.0266 |
| RDN | *Thera Air* | 0.0300 |
| | *Thera Plus* | 0.0360 |
| | *Thera Pro* | 0.0293 |
| Theoretical Gaussian | | 0.0702 |

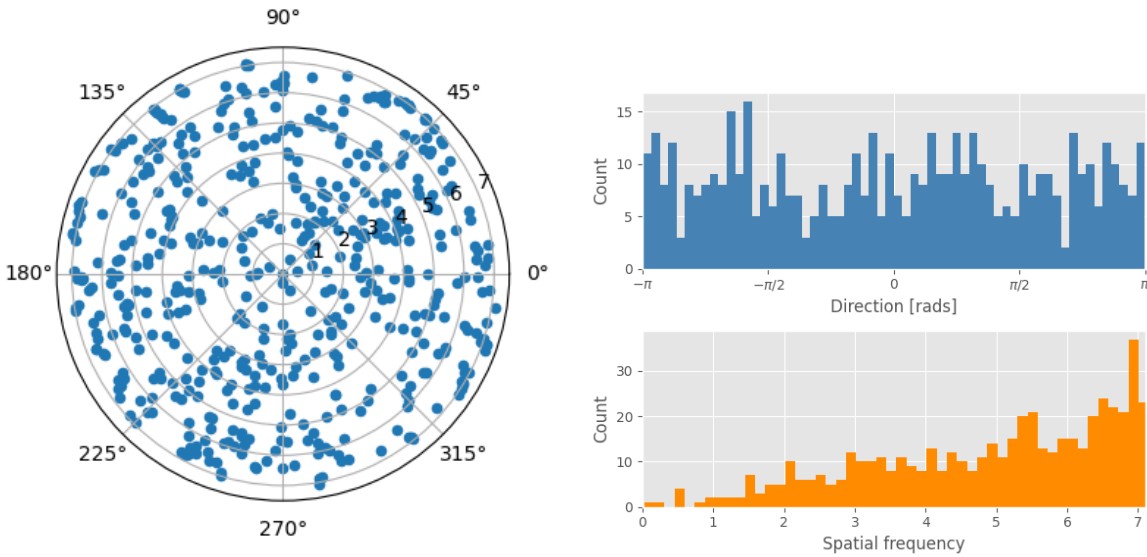

Figure 11: Statistical distribution of the converged frequency bank of the *Thera Pro* run with an RDN backbone. *Left:* Polar scatter plot of spatial frequency and angular direction of components. *Right:* Corresponding marginal distributions.

# E Hypernetwork Architecture

In our implementation, *Thera Air* uses no feature refinement blocks on top of the backbone except a $1 \times 1$ convolution mapping pixel-wise features into field parameters, as done in SIREN (Sitzmann et al., 2020b). *Thera Plus* uses 6 ConvNeXt (Liu et al., 2022) blocks with $d = 64$ followed by 7 ConvNeXt blocks with $d = 96$ and 3 ConvNeXt blocks with $d = 128$, prior to the final mapping layer. Projection blocks are added between blocks with different $d$, which consist of a layer normalization and a $1 \times 1$ convolution operation. For *Thera Pro*, two windowed transformer-based blocks (Liang et al., 2021; Liu et al., 2021) blocks are used with 7 and 6 layers respectively, and 6 attention heads in each layer.

There are 128 field parameters for *Thera Air* and 2048 for the larger variants *Thera Plus* and *Thera Pro*. These numbers are computed as follows: Let $c$ be the number of components used (32, 512, and 512 respectively), then we need $c$ field parameters indicating phase shift and $3c$ field parameters for the linear mapping between components and RGB channels, resulting in a total of $4c$ field parameters produced by the hypernetwork. The total parameter count of the hypernetwork is 8,192, 1.41 M, and 4.63 M for the three variants, respectively. To this we have to add the parameters of the components themselves (*i.e.*, the mapping from coordinate space to thermal activation arguments defined by $\mathbf{W}_1$) that are defined in a global, learnable frequency bank, adding further $2c$ parameters. We highlight that these do **not** come from the hypernetwork.

Table 10: Comparison of runtime and VRAM footprint of various methods with an EDSR-baseline backbone.

| Method | Add. params. | Time | VRAM |
|--------|-------------|------|------|
| MetaSR | 0.45 M | 3.72 ms | 834 MiB |
| LIIF | 0.35 M | 13.8 ms | 604 MiB |
| LTE | 0.49 M | 16.7 ms | 574 MiB |
| CiaoSR | 1.43 M | 37.5 ms | 3894 MiB |
| CLIT | 15.7 M | 86.5 ms | 9402 MiB |
| SRNO | 0.80 M | 11.2 ms | 2702 MiB |
| MSIT | 4.83 M | 92.2 ms | 9388 MiB |
| *Thera Air* | .008 M | 3.56 ms | 322 MiB |
| *Thera Plus* | 1.41 M | 14.2 ms | 664 MiB |
| *Thera Pro* | 4.63 M | 19.44 ms | 686 MiB |

## F   Computational Complexity

Table 10 reports inference time and VRAM requirements of different Thera variants as well as comparison methods, on the $48 \times 48$ pixels standard input patch size and scaling factor 4 (i.e., $192 \times 192$ output size). Each Thera variant improves compute time and memory efficiency compared to methods with similar or higher parameter counts, with particularly pronounced gains over Transformer-based competitors (*e.g.*, Thera Pro uses less than 1/4 of the time and 1/13 of the VRAM footprint of MSIT, at similar parameter count). All tests were performed on an NVIDIA GeForce RTX 3090 Ti GPU.

## G   Real-World Optical Zoom Data

To evaluate the effectiveness of our approach on real-world continuous optical zoom data, we conducted experiments using the COZ dataset (Fu et al., 2024). We introduce *Thera++*, which combines two components: (1) *Thera Plus* (EDSR) trained on the COZ dataset, and (2) a lightweight spatial transformer network (STN) (Jaderberg et al., 2015) that estimates just 6 parameters per image to correct domain-specific affine distortions. Despite its name, the STN is not a transformer network in the modern sense that employs attention mechanisms; rather, it is an image model block that explicitly allows the spatial manipulation of data within a convolutional neural network.

The STN component follows a standard architecture, consisting of a simple convolutional localization network with adaptive pooling to handle variable input sizes. The network processes the input image along with the scale factor and outputs six parameters of a 2D affine transformation ($xy$-translation, anisotropic $xy$-scale, rotation and shear). With approximately 10K parameters, this lightweight component efficiently corrects for geometric distortions while adding minimal computational overhead.

Table 11: Results (PSNR in dB) on the COZ test set.

| Method | In-distribution | | | Out-of-distribution | |
|--------|------|------|------|------|------|
| | $\times 2$ | $\times 3$ | $\times 4$ | $\times 5$ | $\times 6$ |
| MetaSR | 28.70 | 26.55 | 25.17 | 24.31 | 23.25 |
| LIIF | 28.72 | 26.61 | 25.16 | 24.32 | 23.23 |
| LTE | 28.67 | 26.55 | 25.15 | 24.37 | 23.26 |
| SRNO | 28.73 | 26.59 | 25.15 | 24.31 | 23.25 |
| LIT | 28.74 | 26.58 | 25.15 | 24.35 | 23.19 |
| LMI | 28.86 | 26.66 | 25.22 | 24.39 | 23.29 |
| Thera++ | **29.06** | **26.84** | **25.45** | **24.49** | **23.46** |

*Thera++* addresses an important limitation of the COZ dataset. When looking at the data it becomes obvious that COZ samples are not perfectly aligned, resulting in *xy*-jitter between images. Additionally, dynamic objects like people, dust, leaves, and moving shadows appear inconsistently across images of the same scene, significantly increasing the noise level. These challenges make super-resolution particularly difficult for this dataset. However, *Thera++* outperforms previous state-of-the-art methods, including LMI, across all scaling factors, highlighting its applicability under real-world imaging conditions, see Table 11.

## H    Reproducibility of Existing Methods

We encountered challenges attempting to recreate the results reported for some of the competing methods, which explains why some are missing or differ from the originally reported numbers. Details are provided below.

**CUF (Vasconcelos et al., 2023).** At the time of writing, there were no public code or checkpoints available for CUF. Therefore, we could not generate numbers for datasets and scaling factors not reported in the original paper. We have denoted those missing values with "—" in the tables. Furthermore, we could not create any qualitative samples using CUF.

**CLIT (Chen et al., 2023).** For CLIT, at the time of writing, there is a public code, but no checkpoints. We have made a bona fide attempt to reproduce the models, but due to the cascaded training schedule and the large model size, the training process would require excessive amounts of compute: over a month using $8\times$ Nvidia GeForce RTX 3090 GPUs. Unfortunately, the authors did not respond to our requests for the trained checkpoints used in their paper.

**CiaoSR (Cao et al., 2023).** We found that in the official CiaoSR implementation border cropping prior to evaluating on DIV2K deviated slightly from all other methods. We thus adapted the evaluation code to match the competition and enable a meaningful comparison. All DIV2K numbers were re-computed with the corrected code, resulting in slight deviations from those reported in the original paper.

**MSIT (Zhu et al., 2025).** The numbers reported in the paper were achieved by training on a roughly $3\times$ larger training set, comprising not only DIV2K but also Flickr2K. For a fair comparison with all other ASR methods used in our paper, we re-trained MSIT on DIV2K alone. We used the authors' official code and configuration files and trained for two stages, each comprising 1050 epochs. For the second (RiM) training stage we used scaling factors between $\times1$ and $\times4$. Performance generally drops as a result of the smaller training set, in line with the ablation experiments reported for MSIT, where the models were also trained only with DIV2K.

