# OpenReview forum: "Thera: Aliasing-Free Arbitrary-Scale Super-Resolution with Neural Heat Fields"
_TMLR — Accepted by TMLR_

### Review · Reviewer_jQXV · 2025-08-03

**Summary Of Contributions:**

The paper proposes a bunch of anti-aliasing techniques for arbitrary super resolution tasks. These include 1) a learnable Fourier feature mapping, where the frequency 'bank' is learnable; 2) multi-scaled training data is used for training the prior model; 3) band-limited activation function; 4) TV regularization to encourage smoothness.

The paper is clear in general, and there are some valid justification behind how to set the hyperparameters especially in the newly proposed activation function.

**Audience:**

Yes

**Audience Explanation:**

The newly proposed activation function has been empirically proved to have reasonable performance in ASR task. The discussion on solving aliasing in INR type approach is valuable to the community.

**Broader Impact Concerns:**

Can this approach be modified a bit and adapt to other reconstruction or inverse problems? Can some of the ideas be useful to the other inverse problems?

**Claims And Evidence:**

No

**Claims Explanation:**

I am concerned with several issues in the paper:

1. What are the finally learned frequency bank? Are they separated with each other in frequency domain? Are they highly dependent on the training data, and how generalizable is the learned frequency bank, can they be used for other reconstruction tasks?

2. Does lower number of model parameters directly reflect on the reduced inference time of the model? Please justify more on this, and more practically, in terms of the running time/complexity.

3. What are the exactly values of $\kappa$ and $t$ used for each upsampling task? If $\kappa$ is trainable, then what is the trained value, is it consistent with the justification made on the value of $\kappa$? The selection of $t$ in practical experiments is not clear to me.

4. The TV regularization can smooth the image, besides PSNR, for some results, can the authors provide SSIM and GMSD to evaluate the final reconstructed image as well?

5. There are several other approaches can be included for comparison in Figure 5, for example SIREN and its variants.

**Requested Changes:**

The concerns are the corresponding requested changes or explanation are in previous sections.

---

> ### Author Response · Authors · 2025-09-03
> **Response from authors**
>
> We appreciate the reviewer's valuable comments and suggestions. We are also happy that the reviewer found our contributions interesting to the community. Below, we address the questions raised.
>
> ## Analysis of learned components
> We would like to point out that Figs. 1 & 3 already depict subsets of the components (RDN-Pro run). Fig. 11 now shows all 512 learned components in polar coordinates, with marginal distributions of direction and spatial frequency. The components are distributed uniformly across directions, with fewer at low and more at high frequencies, providing more capacity to reconstruct fine details - consistent with the intuition that higher-resolution images have more degrees of freedom.
>
> ## Generalizability of learned components
> An interesting question. For ASR, we have already shown Thera’s ability to generalize to OOD datasets. The uniform distribution of components in Fig. 11 further suggests they form a general-purpose frequency basis. To verify this, we fit heat fields to OOD datasets, including even fastMRI [1] (single-coil knee validation split, scaled to [0,1]), fixing the Fig. 11 frequency bank and optimizing only scale/shift (lr = 0.001, AdamW, 600 iters). The new Tab. 8 shows that the pre-trained components achieve near-zero reconstruction MSE for smaller fields (in-distribution) and still low errors for larger (OOD) fields up to $18 \times 18$ pixels.
>
> This is not surprising: the frequency bank was optimized for local fields, and can act as an over-parametrized dictionary for smaller field sizes (e.g., $4\times4$ GT pixels per field). Notably, it still works well even at large OOD sampling factors, and shows no obvious difference between natural and MRI images, suggesting it functions as a general-purpose basis akin to Fourier or DCT. See Appendix D for details.
>
> ## Inference time & memory
> Tab. 10 now reports time and VRAM requirements - both scale with parameter count, with two key observations: (1) All Thera variants are more time- and memory-efficient than methods with similar or larger parameter counts, while performing better; (2) Efficiency gains are especially strong against Transformer-based methods (e.g., Thera Pro uses $<1/4$ of the time and $<1/13$ of the VRAM of MSIT, at similar parameter count; see also Thera Plus vs. CiaoSR).
>
> ## Values of $\kappa$ and $t$
> Thanks for the pointer, we agree the role of $t$ needed clarification, as reviewer ZWRi also noted. In our formulation, $t$ is determined from the desired scaling factor $s$ via Eq. 4 as $t=s^{-2}$ (e.g., $t=0.25$ for $\times2$, $t=0.\overline{1}$ for $\times3$, etc.). It acts as a third continuous field coordinate controlling the Gaussian blur strength, so that the signal can be observed at a given sampling rate without aliasing. Compared to others, this mechanism is analytical and efficient; please also see Appendix A.3 discussing the relationship between $t$ and the subsampling rate. We have improved the writing in Secs. 1 & 3.1.
>
> $\kappa$ is the global thermal diffusivity coefficient, derived so that the Gaussian blur’s cutoff matched the Nyquist frequency at native resolution when $t=1$ (Appendix A.2). In practice it deviates with the AA filter; following LIIF (and all comparisons) we used BC-splines with a cutoff $\sim2\times$ Nyquist to preserve more details. As shown in Tab. 9 & Sec. D, $\kappa$ converges stably to $0.0308 \pm 0.003$, about half the theoretical value, showing that Thera adapts robustly to different filters with a single parameter.
>
> ## Additional metrics
> SSIM scores are already listed in Tab. 4 (Appendix), showing favorable performance of our method. We have now computed GMSD metrics (Tab. 5, with std. devs.), as suggested by the reviewer. We reach significantly lower GMSD for all backbone-scale combinations, indicating more faithful gradient structure. We appreciate the suggestion.
>
> ## Further approaches in Fig. 5
> We had not included SIREN in Fig. 5 since it is not an SR method: it simply (over)fits a field to the data rather than recovering additional details. In contrast, the aim of Fig. 5 was to compare our SR method with several ReLU-based SR competitors, highlighting differences in (higher-order) gradient structure. That said, we have fit a SIREN to the ground-truth image from Fig. 5. As expected, it recovers gradients faithfully. Indeed, SIREN partly motivated our approach, as discussed for Fig. 5.
>
> ## Broader Impact
> Thank you for the insightful question. Indeed, neural heat fields provide a general formulation for multi-scale INRs beyond (A)SR. In future work, they could be useful for other inverse problems where Gaussian signal observations are concerned, e.g., in 3D (anti-aliased NeRFs [2], SDFs, etc.), video, or audio.
>
> ### References
> [1] Zbontar, Jure, et al. "fastMRI: An open dataset and benchmarks for accelerated MRI." arXiv preprint 1811.08839. 2018.
>
> [2] Barron, Jonathan T., et al. "Mip-NeRF: A multiscale representation for anti-aliasing neural radiance fields." CVPR 2021.

---

### Review · Reviewer_ZWRi · 2025-08-04

**Summary Of Contributions:**

Thera: Aliasing-Free Arbitrary-Scale Super-Resolution
with Neural Heat Fields

This paper works on arbitrary-scale super-resolution (SR) with neural heat fields. Previous methods approximate the integral version of the field at each scaling factor. This work introduces a neural heat field that can achieve analytically correct anti-aliasing at any scaling factor.
This paper introduces an interesting formulation of the problem with neural heat fields by using a two-layer perception to parameterize the field. For each pixel, the phase shift b1 and matrix W2 are modeled by the output of a hypernetwork with image prior.
The prior hyper-network can be modeled with different architectures, the authors introduced three kinds of architectures with different capacities, all exhibiting competitive performance.
The authors also conducted quantitative evaluations to show the effectiveness of the proposed method and ablative studies to verify the desigin choices.

Strengths:

1. The proposed method is interesting and verified by the experiments. Although it limits the network to a two-layer perception, the decoupling of frequency bank, phase shift and global bias makes sense.
2. The authors presented multiple architectures for the prior hyper-network, with different number of parameters and capacities, allowing it for real-world use and different purposes.
3. The Thera Air has a simple and efficient architecture, while still have competitive performance compared to previous methods. The authors also conducted ablative studies to verify the design choices.

Limitation:

1. No error bars in the quantitative evaluations, making it hard to understand the marginal performance gain on some of the results.
2. The author argued that previous methods approximate an integral version of the field at each scaling factor. It would be beneficial to lay out what approximations are used in the previous methods and how the proposed method is different in a general framework in the appendix. This will be helpful for the readers in a broader community without referring to other papers.
3. Writing can be improved in some sections. For example, the authors introduced the time coordinate $t$ in section 3.1 abruptly without a clear explanation. In the reader's understanding, it's linked with the scaling factor, but it was not directly explained in the main paper.
4. Although the qualitative studies show the effectiveness of the proposed method, it is still a bit off compared to the ground truth. One might wonder how denoising diffusion models would perform in this case. Would a better prior be more helpful in this case?

Overall, the paper is presented in a good manner, and the proposed method is supported by its experiments. The reviewer believes the strengths outweigh the limitations once the questions and limitations are addressed.

**Audience:**

Yes

**Audience Explanation:**

The proposed method would be interesting to the single-image super-resolution community. The formulation of the continuous representation from point spread functions would also be helpful to other communities, such as neuromorphic computing.

**Broader Impact Concerns:**

The reviewer did not notice significant impact concerns of the proposed method as long as the training data of the proposed method follows the rules and regulations on intellectual properties and privacy.

**Claims And Evidence:**

Yes

**Claims Explanation:**

The authors performed a quantitative study on multiple datasets against previous methods, and the Thera Pro model achieved the best overall performance.

**Requested Changes:**

Most of them have been covered in the limitations part of the review. The error bars and comprehensive analysis on the approximation would be preferred. Section 3.1 could possibly be improved for a smooth introduction. The discussions and analyses about diffusion models would be interesting to readers, too.

---

> ### Author Response · Authors · 2025-09-03
> **Response from authors**
>
> Thank you very much for the thoughtful feedback. We are glad you found our submission interesting to the community and well supported by experiments. We address the limitations pointed out below:
>
> ## Error bars
> Thank you for suggesting this, we have now ran all training runs $N=3$ times and report PSNR, SSIM and GMSD standard deviations in Tables 4, 5 and 7 in the Appendix. We observe very low standard deviation across runs initialized with different random seeds, e.g., on DIV2K < 0.004 dB PSNR for Thera-Pro with RDN backbone and < 0.015 dB with the lighter EDSR backbone.
>
> These standard deviations are significantly smaller than the reported performance gains.
>
> ## Approximations in previous approaches
> Previous approaches (LIIF, CiaoSR, MSIT, CLIT, etc.) provide the scaling factor - or, equivalently, the area of the output pixels - as additional input to the neural field. The MLP then learns from the training data to output an appropriately blurred (i.e., anti-aliased) field according to the scaling factor, suppressing higher frequencies for lower scaling factors, thereby approximating the observation model.
>
> Our heat fields also receive an additional coordinate, namely the "time" $t$, but the resulting filtering operation is not learned from data, rather it is expressed analytically by thermal activations. In other words, other approaches fit a 3D field ($x$, $y$, $scale$) while we only need to fit a 2D field ($x$, $y$), whereas the scale "dimension" is computed analytically; resulting in improved data efficiency. We have made this clearer in the corresponding paragraph in the introduction.
>
> ## Writing
> Thank you for the pointer. In our formulation, $t$ acts as a third, continuous input coordinate to the neural field and controls the strength of the Gaussian blur applied by the thermal activations (see previous answer). Its value follows directly from the scaling factor at which the field should be observed (see Eq. 4 in the paper for the exact mapping between the **sub**sampling rate and $t$). We named it $t$ since it corresponds to time in the heat equation that served as inspiration - as time passes, the signal is increasingly blurred. Section A.3 in the appendix describes in detail the connection between $t$ and the scaling factor. We have improved the corresponding text in Secs. 1 and 3.1 to clarify the role of this variable.
>
> ## Discussion on Denoising Diffusion Models
> We appreciate the reviewer's suggestion and agree that a better prior (like the one in diffusion models) can improve perceptual quality. However, it is important to recognize that this corresponds to a different objective. Generative models are trained to produce perceptually convincing outputs by predicting one of many *plausible* high-resolution images. But as a consequence they introduce more distortion (resulting on worse PSNR and SSIM), since the true ground truth details are not exactly recovered.
>
> By contrast, our approach - and also comparable methods with the same goal - minimizes per-pixel errors. I.e., the target is an MMSE estimate, with higher PSNR/SSIM and free of hallucinations, at the cost of slightly smoothed appearance due to the "regression to the mean" effect [1]. The inevitable balancing between sharpness and fidelity is known as the perception-distortion tradeoff [2]; Figure 5 in [1] shows this tradeoff in practice when using generative methods for super-resolution. In the present work we aim for faithfulness to the ground truth, as required for many downstream applications based on the super-resolved images.
>
> Directly comparing our method to diffusion models with the same metrics would be inappropriate, they are known to reach significantly lower PSNR in exchange for higher perceptual quality. E.g., a recent diffusion-based ASR method, IDM [3] reports a PSNR of 27.59 dB on the $\times4$ DIV2K upsampling task, much lower than LIIF's 29.00 (SSIM: 0.78 (IDM) vs. 0.89 (LIIF)).
>
> We have clarified this distinction in the related work section, referencing recent diffusion SR works and emphasizing that while they generate impressively realistic images, they are evaluated on different criteria and generally achieve lower scores on distortion metrics.
>
> ### References
>
> [1] Delbracio, Mauricio, and Peyman Milanfar. "Inversion by Direct Iteration: An Alternative to Denoising Diffusion for Image Restoration." Transactions on Machine Learning Research.
>
> [2] Blau, Yochai, and Tomer Michaeli. "The perception-distortion tradeoff." Proceedings of the IEEE conference on computer vision and pattern recognition. 2018.
>
> [3] Gao, Sicheng, et al. "Implicit diffusion models for continuous super-resolution." Proceedings of the IEEE/CVF conference on computer vision and pattern recognition. 2023.

---

### Review · Reviewer_HwDs · 2025-08-24

**Summary Of Contributions:**

This paper presents a novel approach to overcoming aliasing in single-image any scale superresolution (ASR). First they introduce "neural heat fields", a new type of neural field grounded in signal processing theory that garuntees anti-aliasing by design. These neural heat fields analytically model the physical point spread function (PSF) of images. The neural heat field are then incorporated in a novel end-to-end ASR method called "Thera" that combines a hypernetwork with a grid of local neural fields. Experiments show that compared to other ASR methods Thera outperforms while being more parameter-efficient.

### Strengths
- The paper incorporates a few principled ideas from traditional image processing such as neural heat fields and TV regularization. This makes their method more interpretable and could provide some insights into how we design general neural fields.
- The authors demonstrate 3 variants of Thera making it customizable for a range of computing resources.
- The results are very convincing showing that Thera acheives a good or better PSNR than many other methods across a range of datasets.

### Weaknesses
- By design, neural heat fields only allow for a single hidden layer MLP representation. While this is sufficient for the tasks studied, it may limit the expressive capacity of the model compared to deeper neural fields. The authors note this and compensate with a heavier hypernetwork, but this restriction could hinder extension to more complex settings.
- While the authors make $\kappa$ trainable, its role and learned values are analyzed. Understanding how $\kappa$ adapts in practice could provide insight into whether the model is truly learning a physically meaningful observation model or just tuning a free parameter.

**Audience:**

Yes

**Audience Explanation:**

Neural fields are very popular in the TMLR and machine learning community so I see this paper being of interest to much of TMLR's audience.

**Broader Impact Concerns:**

None.

**Claims And Evidence:**

Yes

**Claims Explanation:**

Yes all claims made in the paper are accurate and convincing. The theoretical basis for neural fields is well explained and convincing. Moreover, the authors provide extensive experiments and compare across many other ASR methods backing up their claims that Thera is a significantly better model for ASR and that it offers significant savings in terms of parameters.

**Requested Changes:**

- Provide more discussion or visualization of the learned $\kappa$ values across datasets and scales. This would clarify whether the parameter is learning meaningful physical structure.
- This is more of a suggestions, but it would be interesting to see whether this approach can be useful for other neural field tasks such as computed tomography reconstruction with few samples.

---

> ### Author Response · Authors · 2025-09-03
> **Response from authors**
>
> We thank the reviewer for their thoughtful comments and positive view of our work. We address below the open questions that were pointed out in the review.
>
> ## Single hidden layer MLP
> We agree that the restriction to a single hidden layer limits the expressivity of the field, as discussed in the manuscript (Sec. 4.3), but this was the only way we were able to formulate the field blur operations analytically. Theoretical proofs concerning deeper networks are notoriously complicated [1]. Using a hypernetwork largely circumvents this limitation, as shown by our results, but it does come with a complexity overhead. We would be delighted if our work was extended to multilayer architectures in the future.
>
> ## Analysis of $\kappa$
> An analysis of $\kappa$ is indeed a good idea and was also requested by reviewer jQXV. In the Appendix (Sec. D) we have therefore added an analysis of learned values for $\kappa$. In summary, what we observed was that across runs, $\kappa$ converged very consistently to $0.0308 \pm 0.003$. Please note that by nature, the value of $\kappa$ is shared across scales for each run.
>
> The observed deviation from the theoretical value can be explained: In our implementation, we followed the data generation protocol laid out in LIIF (and adapted by all comparison methods), a more practical bicubic spline from the Mitchell-Netravali filter family [2] as implemented in the PIL library. In effect, we see our model learning a value for $\kappa$ such that its Gaussian observation model best approximates the actual filter applied during training. This demonstrates that, despite being derived from a theoretical Gaussian model, Thera can adapt seamlessly to practical approximations by tuning a single parameter.
>
> ## Other neural field tasks
> We appreciate the reviewer’s suggestion to explore neural heat fields for CT reconstruction (or even super-resolution) from sparse data. We believe this is a promising research direction that would warrant a dedicated study of its own. In future work, the component attenuation enabled by the thermal activations could even be modified to be direction-dependent, which might be beneficial for this type of reconstruction.
>
> ### References
>
> [1] Hornik, Kurt, Maxwell Stinchcombe, and Halbert White. "Multilayer feedforward networks are universal approximators." Neural networks 2.5 (1989): 359-366.
>
> [2] Mitchell, Don P., and Arun N. Netravali. "Reconstruction filters in computer-graphics." ACM Siggraph Computer Graphics 22.4. 1988.

---

### Decision · Action_Editor_Pwtm · 2025-10-12

**Recommendation:** Accept as is

**Audience:**

Yes

**Audience Explanation:**

This paper studies image super-resolution, a key inverse problem. Reviewers believe this is a valuable contribution to TMLR, and I agree.

**Claims And Evidence:**

Yes

**Claims Explanation:**

This paper introduces a new approach for single image super-resolution based on neural heat fields, modeling the physical point spread function and allowing for analytically anti-aliasing super-res at any scale. Unlike other methods (like implicit neural representations) that approximate the observation integral, the proposed method finds an analytical expression for it at no additional computational cost. The method and claims are well supported by the presented results.

All reviewers are supportive of the level of contributions in this paper. Furthermore, their questions and suggestions have led to further improvements in the paper, including statistical details in experiments, clarifications on $t$ and $\kappa$, and limitation of the modeling via a single MLP.

All reviewers agree that this is a valuable contribution to the problem of single image super-resolution, and I agree.